# Variables That Could Influence Healing Time in Patients with Diabetic Foot Osteomyelitis

**DOI:** 10.3390/jcm12010345

**Published:** 2023-01-01

**Authors:** Aroa Tardáguila-García, Francisco Javier Álvaro-Afonso, Marta García-Madrid, Mateo López-Moral, Irene Sanz-Corbalán, José Luis Lázaro-Martínez

**Affiliations:** 1Diabetic Foot Unit, Clínica Universitaria de Podología, Facultad de Enfermería, Fisioterapia y Podología, Universidad Complutense de Madrid, 28040 Madrid, Spain; 2Instituto de Investigación Sanitaria del Hospital Clínico San Carlos (IdISSC), 28040 Madrid, Spain

**Keywords:** diabetic foot osteomyelitis, healing time, signs of infection, blood parameters, radiological signs, medical treatment, surgical treatment

## Abstract

Aim: To compare the healing time in patients with diabetic foot osteomyelitis according to the presence or absence of clinical signs of infection, variation of blood parameters, the presence of different radiological signs, and the treatment received for the management of osteomyelitis. Methods: A prospective observational study was carried out in a specialised Diabetic Foot Unit between November 2014 and November 2018. A total of 116 patients with osteomyelitis were included in the study (treated by either a surgical or medical approach). During the baseline visit, we assessed the diagnosis of osteomyelitis, demographic characteristics and medical history, vascular and neurological examination, clinical signs of infection, increased blood parameters, radiological signs of osteomyelitis, and the treatment to manage osteomyelitis. We analysed the association between the presence of clinical signs of infection, variation of blood parameters, presence of radiological signs, and treatment received for management of osteomyelitis with the healing time. Results: The mean time to ulcer healing was 15.8 ± 9.7 weeks. Concerning healing times, we did not find an association with the presence of clinical signs of infection or with the increase in blood parameters, except in the case of eosinophils, which with higher values appear to increase the healing time (U = 66, z = −2.880, *p* = 0.004). Likewise, no relationship has been found between healing time and the appearance of the different radiological signs of osteomyelitis, nor depending on the treatment administered for the management of osteomyelitis. Conclusion: High levels of eosinophils are associated with a longer healing time of diabetic foot ulcers complicated with osteomyelitis, finding no other factors related to increased healing time.

## 1. Introduction

Diabetic foot osteomyelitis (DFO) is considered a moderate to severe infection, which can lead to major complications, such as loss of the lower limb or death of the patient [1].

Its diagnosis and treatment have been studied over the years, confirming the importance of an early diagnosis and multidisciplinary management directed to meet the needs of each case [1,2,3]. However, an important aspect in the follow-up of the patient suffering from osteomyelitis is to know the possible risk factors that may influence the patient’s recovery.

We know that DFO cannot be cured until after 12 months and that it should be considered osteomyelitis in remission [4]. In addition, the short- and long-term complications [5] of patients with DFO have been studied, regardless of the treatment by which they have been managed (medical or surgical) [4]. We have also looked for cure criteria during these 12 months through complementary tests, such as blood tests or X-rays [6,7].

Nevertheless, whether certain factors may lead us to suspect that the healing of a patient with DFO may be delayed has not been described. For this reason, here we compared the time until healing in patients who suffer from DFO according to the presence of clinical signs of infection, the increased blood parameters, the presence of different radiological signs, and according to the treatment received for the management of DFO.

## 2. Methods

A prospective observational study was carried out in a specialised Diabetic Foot Unit between November 2014 and November 2018. A total of 116 patients with DFO were included in the study.

Inclusion criteria were as follows: patients with DFU complicated with DFO; patients > 18 years; patients who demonstrated healing after receiving the management of DFO; and patients who agreed to be included after signing written consent. Exclusion criteria were patients who required major amputation to treat DFO; patients who required revascularisation and patients with critical limb ischemia according to the IWGDF 2019 guidelines [8]; pregnant or lactating women.

During baseline visits, we assessed the following items of the patients included in the study: diagnosis of DFO, demographic characteristics and medical history, vascular and neurological examination, presence of clinical signs of infection, increased blood parameters, radiological signs, and treatment to manage DFO.

DFO was diagnosed first by clinical suspicion of bone infection through the probe-to-bone test (PTB) and simple X-ray [9]. The PTB test was positive when the researcher felt a hard or gritty surface through the ulcer using metal forceps (Halsted mosquito forceps). Two standard views of X-ray were performed, and we considered the X-ray positive when the research could visualise the following radiological signs in X-ray: focal loss of trabecular pattern or marrow radiolucency (demineralisation); periosteal reaction or elevation; the presence of sequestrum (devitalised bone with radiodense appearance separated from normal bone); loss of bone cortex, with bony erosion or demineralisation; and other types of signs (bone sclerosis, with or without erosion; the presence of involucrum (layer of new bone growth outside previously existing bone resulting and originating from stripping off the periosteum), and the presence of cloacae (opening in the involucrum or cortex through which sequestrum or granulation tissue may discharge)) [10]. Subsequently, DFO diagnosis was confirmed by positive bone culture or positive histology result [11].

The vascular examination consisted of palpation of the distal pulses (dorsal pedis and posterior tibial); completion of the ankle-brachial index (ABI); completion of the toe-brachial index (TBI); and assessment of transcutaneous oxygen pressure (TcPO2) (TCM4 transcutaneous monitor; Radiometer Medical, Brønshøj, Denmark) [8]. A peripheral arterial disease was considered when the patient presented an absence of palpation of distal pulses and ABI < 0.9. Further, PAD was considered when the patient presented an absence of palpation of distal pulses and ABI > 1.4 (medial arterial calcification), with the TBI < 0.7 and TcPO2 < 30 mmHg [12,13].

The neurological examination consisted of the assessment of superficial sensitivity using a Semmes-Weinstein 5.07/10 g monofilament (Novalab Ibérica, Alcal. de Henares, Madrid, Spain) and the assessment of deep sensitivity using a tensiometer (Me.Te.Da. S.r.l., San Benedetto del Tronto, Italy) [14].

The presence or absence of clinical signs of infection was assessed by the same investigator (A.T.-G.). On the one hand, the classic clinical signs of infection were assessed: pain, flushing, warmth, and swelling. On the other hand, the signs of infection proposed by the Infectious Diseases Society of America (IDSA) guidelines [15] were also assessed: evidence of systemic inflammatory response, rapid progression of infection, extensive necrosis or gangrene, crepitus on examination or soft tissue gas on imaging, extensive ecchymosis or petechiae, especially haemorrhagic bullae, new onset wound, pain out of proportion to clinical findings, the recent loss of neurological function, critical limb ischaemia, extensive soft tissue loss, extensive bony destruction especially in midfoot and hindfoot, and failure of infection to improve with appropriate therapy.

Laboratory blood tests were performed in all patients, within 24 h or up to a maximum of 48 h after diagnosis of DFO, considering the elevated blood parameters as shown below: leukocytes > 11 × 10^9^/L; neutrophils > 6.8 × 10^9^/L; lymphocytes > 3.7 × 10^9^/L; monocytes > 1.1 × 10^9^/L; eosinophils > 0.5 × 10^9^/L; basophils > 0.1 × 10^9^/L; erythrocyte sedimentation rate (ESR) > 20 mm/h; glycaemia > 5.6 mmol/L; glycosylated haemoglobin (HbA1c) > 7.7 mmol/L; C-reactive protein (CRP) > 476.2 nmol/L; alkaline phosphatase > 129 UI/L; albumin > 48 g/L; and creatinine > 1.1 mg/dL.

Regarding the X-ray assessment, we grouped the presence of radiological signs compatible with DFO as follows: affected bone marrow, active periosteal reaction, sequestrum, cortical disruption, and other types of signs. All X-rays were evaluated by the same researcher (A.T.-G.).

Management of DFO was by surgical or medical treatment, following previously published recommendations [16]. All the surgeries were performed by the same surgeon (J.L.L.-M.), a specialist in conservative foot surgery, defined as procedures in which only infected bone and non-viable soft tissue are removed, but no amputation of any part of the foot is undertaken [17]. Patients who were managed with medical treatment first received empirical antibiotics, following the recommendation of international guidelines [15], and then the treatment was modified according to the result of the bone culture [18]. Antibiotic treatment was maintained for 6 weeks [19].

The ulcer was considered healed when there was no drainage at the ulcer site 2 weeks after complete epithelialisation [20]. All patients received twice-weekly assessments until the ulcer healed. According to international guideline recommendations, local ulcer dressings consist of dressing selection based on exudate control, comfort, and cost [21].

The study was carried out and completed according to the ethical standards of the responsible committee, and ethical approval was obtained (code: 14/485-E). Informed consent was collected from all patients included in the study. The authors declare that the study complied with the ethics code of the Declaration of Helsinki [22].

Statistical analysis was performed using the SPSS^®^ statistical software version 20.0 for iOS (SPSS, Inc. Chicago, IL, USA). The sample description has been presented by expressing qualitative variables as frequency (n) and percentage (%) and quantitative variables as mean and standard deviation. The Kolmogorov–Smirnov normality test was performed for all quantitative variables used in this study, and none showed signs of being normally distributed. For the comparison of quantitative variables between the two groups, the Mann–Whitney U test was used. The Kaplan–Meier test was used to describe the survival of healing time in patients with elevated eosinophil values. Differences were considered significant at *p* < 0.05 for a confidence interval of 95%. The sample size was calculated using GRANMO, sample size calculation public software version 7.12, estimating a required sample of 80 subjects (study power—80.0%, type 1 error—5.0%, and predictable loss to follow-up—15.0%).

## 3. Results

One-hundred-sixteen patients with DFO were included in the study. The mean time for ulcer healing was 15.8 ± 9.7 weeks. The demographic and basic clinical characteristics of the sample are summarised in Table 1.

According to the presence of clinical signs of infection, we observed that there were no statistically significant differences in healing times between patients who had clinical signs of infection associated with the DFO process and those who did not present clinical signs of infection (13 (IQR = 8–20) versus 16 (IQR = 8–22) weeks, respectively, *p* = 0.550).

Regarding the increasing blood parameters, we observed that patients with increased eosinophil values took longer to heal at 35 (IQR = 23–35) weeks compared to patients with normal values of eosinophil 13 (IQR = 8–20) weeks (U = 66, z = −2.880, *p* = 0.004). Figure 1 shows the Kaplan–Meier survival curve for healing time in patients with elevated eosinophils *p* = 0.005. In the case of basophils, this analysis could not be performed as there were no subjects with increased values, so this variable could not be included in the statistical analyses. The rest of the blood parameters analysed showed no relationship between their increase and longer time to healing (Table 2).

Differences in healing time between the two groups resulting from the presence or absence of each of the radiological signs of osteomyelitis observed were also analysed. No significant differences in healing time were observed in the groups where the different radiological signs analysed were present or absent: affected bone marrow (11.5 (IQR = 8–17) versus 14 (IQR = 8–22) weeks) (*p* = 0.194); active periosteal reaction (12 (IQR = 8–17) versus 13 (IQR = 8.5–22) weeks) (*p* = 0.232); sequestrum (23 (IQR = 7.5–26.5) versus 13 (IQR = 8–20) weeks) (*p* = 0.238); cortical disruption (20 (IQR = 10–23) versus 12 (IQR = 8–20) weeks) (*p* = 0.158); and other types of signs (17.5 (IQR = 10–22.5) versus 13 (IQR = 8–20) weeks) (*p* = 0.438).

Finally, the differences in healing times are also analysed, considering whether the patient had received surgical or medical treatment for the management of osteomyelitis. There were no significant differences between surgical versus medical treatment (13 (IQR = 8.5–21) versus 13 (IQR = 7–22) weeks, respectively) (*p* = 0.720).

## 4. Discussion

Concerning healing times, our results show that they are not associated with the presence of clinical signs of infection nor with the increase in blood parameters, except in the case of eosinophils, which with higher values appear to increase the healing time. Likewise, no relationship has been found between healing time and the appearance of the different radiological signs of DFO, nor depending on the treatment administered for the management of osteomyelitis.

The data obtained regarding healing time are controversial, as we might think that patients with a local or systemic inflammatory response could have a worse prognosis, more complications and, consequently, longer healing time. The authors consider the finding of a relationship between the increase in eosinophils and the increase in time to healing to be a casual and isolated finding, as the other blood inflammatory parameters have not been shown to be related to an increase in healing times, and, furthermore, there is no data in the scientific literature to justify this finding. We might even think more severe radiological signs or more advanced stages could be related to longer healing times. Despite all this, the results seem to show that when a patient receives the treatment indicated for the management of DFO, the response in terms of the healing time is homogeneous, demonstrating that it is more important to plan the application of appropriate treatment for each case than the severity of the infection at the time of diagnosis of the osteomyelitis.

Regarding the relationship between inflammatory markers and healing time in patients with DFO, a paper [23] whose objective was to analyse the predictive role of leukocytes, ESR and CRP, in the healing time of DFO treated by surgery or antibiotics concluded that there is not enough evidence to define the prognostic role of these inflammatory markers in the healing time of ulcers complicated with DFO, regardless of the treatment administered.

A previous study [24], whose objective was to analyse ESR and CRP for monitoring the treatment of DFO, showed that healing is multifactorial, thus justifying the fact that no association was found between the characteristics of the treatment applied at the beginning and in the follow-up after healing. On the other hand, the first clinical trial (carried out by our research group) to compare surgical versus medical treatment in the management of DFO [25], as in our study, also found no significant differences in terms of median healing between the two groups (*p* = 0.72): 6 weeks (IQR = 3–9) in the surgical treatment group and 7 weeks (IQR = 5–8) in the medical treatment group.

The main limitation of our study is that we did not compare our results with a control group (patients without DFO), so the results are only extrapolated to patients who have had DFO. We also find it interesting to mention that the blood extractions performed in the diagnosis of DFO were performed after the start of antibiotic therapy. In addition, another limitation of the study was the difference between the number of patients who received surgical versus medical treatment, with the number of patients managed surgically being higher. In the study, multivariate analysis, including other risk factors, such as ulcer size or the presence of ischaemia, has not been performed.

The main strength is that this is the first study that has evaluated the time until healing in patients who suffer from DFO according to the presence of clinical signs of infection, increased blood parameters, the presence of different radiological signs, and according to the treatment received for the management of DFO.

## 5. Conclusions

High levels of eosinophils are associated with a longer healing time of diabetic foot ulcers complicated with osteomyelitis. Healing times in patients with DFO are not increased despite clinical signs of infection, the presence of different radiological signs of DFO, according to medical or surgical treatment for managing DFO, or increased blood parameters, except eosinophils.

## Figures and Tables

**Figure 1 jcm-12-00345-f001:**
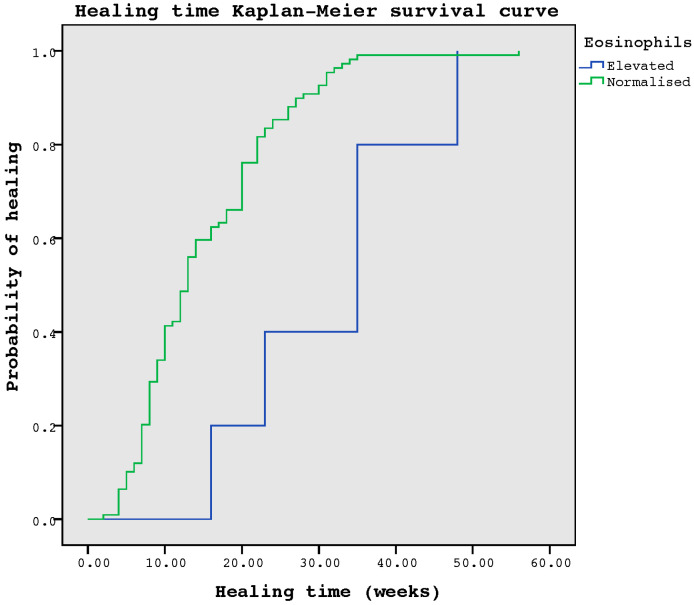
Kaplan–Meier survival curve for healing time in patients with elevated eosinophils.

**Table 1 jcm-12-00345-t001:** Participants’ baseline clinical data.

Variables	*n* = 116
Male/female, *n* (%)	96 (82.2)/20 (17.2)
Age (years), mean ± SD	62.9 ± 10.1
DM (years), mean ± SD	17.5 ± 12.3
DM type 1/type 2, *n* (%)	12 (10.3)/104 (89.7)
Body mass index (Kg/m^2^), mean ± SD	28.3 ± 5.5
HbA1c (%), mean ± SD	8.1 ± 6.0
Neuropathy, *n* (%)	116 (100.0)
Affected superficial sensitivity, *n* (%)	116 (100.0)
Affected deep sensitivity, *n* (%)	116 (100.0)
PAD, *n* (%)	48 (41.4)
ABI; mean ± SD	1.0 ± 0.3
TBI; mean ± SD	0.7 ± 0.2
TcPO_2_; mean ± SD	34.1 ± 13.9
Duration from ulcer (weeks), mean ± SD	15.7 ± 32.1
Location of ulcer: forefoot/midfoot/hindfoot, *n* (%)	107 (92.2)/5 (4.3)/4 (3.4)
Presence of clinical signs of infection, *n* (%)	72 (62.1)
Radiological signs in X-ray: focal loss of trabecular pattern or marrow radiolucency (demineralisation)/periosteal reaction or elevation/sequestrum/loss of bone cortex, with bony erosion or demineralisation/another type of signs, *n* (%)	34 (29.3)/36 (31)/12 (10.3)/26 (22.4)/8 (7)
Blood parameters: leukocytes (×10^9^/L)/neutrophils (×10^9^/L)/lymphocytes (×10^9^/L)/monocytes (×10^9^/L)/eosinophils (×10^9^/L)/basophils (×10^9^/L)/ESR (mm/h)/glycaemia (mmol/L)/CRP) (nmol/L)/alkaline phosphatase (UI/L)/albumin (g/L)/creatinine (mg/dl), mean ± SD	8.7 ± 2.5/5.6 ± 2.3/2.1 ± 0.9/0.7 ± 0.5/0.3 ± 0.2/0.04 ± 0.04/34.3 ± 28.4/8.45 ± 3.16/99.6 ± 46.5/40 ± 6/1.6 ± 1.7
Surgical treatment/medical treatment	96 (82.2)/20 (17.2)

Abbreviations: SD, standard deviation; DM, Diabetes *mellitus*; PAD, Peripheral Arterial Disease; HbA1c, glycated haemoglobin; ABI, ankle-brachial index; TBI, toe-brachial index; TcPO2, transcutaneous oxygen pressure; ERS, erythrocyte sedimentation rate; CRP, C-reactive protein.

**Table 2 jcm-12-00345-t002:** Comparison of healing times in patients with increasing versus normal blood parameters.

Variables	Mean (weeks)	IQR (weeks)	*p*-Value
Elevated leukocytes	13	9–21	0.761
Normalised leukocytes	13	8–22	
Elevated neutrophils	18	12–22	0.117
Normalised neutrophils	12	8–20	
Elevated lymphocytes	8	7.5–8.5	0.057
Normalised lymphocytes	13	8–22	
Elevated monocytes	12	6–20	0.557
Normalised monocytes	13	8–22	
Elevated eosinophils	35	23–35	0.004
Normalised eosinophils	13	8–20	
Elevated ESR	13	9–21	0.378
Normalised ESR	12	8–22	
Elevated glycaemia	12	8–22	0.831
Normalised glycaemia	14	8.5–20	
Elevated HbA1c	13	8–22	0.716
Normalised HbA1c	16	8–20	
Elevated CRP	13	8–22	0.278
Normalised CRP	12	8–20	
Elevated alkaline phosphatase	19	12–23	0.096
Normalised alkaline phosphatase	13	8–20	
Elevated albumin	11	10–12	0.634
Normalised albumin	13	8–22	
Elevated creatinine	12	8–23.5	0.972
Normalised creatinine	13	8.5–20	

Abbreviations: ESR, erythrocyte sedimentation rate; HbA1c, glycosylated haemoglobin; CRP, C-reactive protein.

## Data Availability

The data are available previous request to corresponding author.

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
