# Peer review of "Variables That Could Influence Healing Time in Patients with Diabetic Foot Osteomyelitis"

_jcm, 2023, doi:10.3390/jcm12010345_

Round 1
Reviewer 1 Report
Thank you for the opportunity to review the work. I have no critical comments.I can point out that the study group is characterized by too large a
discrepancy between the subjects 96 male, 20 female, surgical treatment 96,
medical 20 etc. Looking at the literature attached, the question arises whether the analyzed
research material (patients) is the same as in previous publications?
Author Response
Thank you very much for your review. Following we are answering concerns that you have detailed in your review. Yellow highlight text indicates the modified or added text in the new version of the manuscript.
Reviewer: 1
Comments and Suggestions for Authors
Thank you for the opportunity to review the work. I have no critical comments. I can point out that the study group is characterized by too large a discrepancy between the subjects 96 male, 20 female, surgical treatment 96, medical 20 etc. Looking at the literature attached, the question arises whether the analyzed research material (patients) is the same as in previous publications?
Thank you very much for your comment and for taking the time to review.
Regarding the higher percentage of males in the sample, the authors consider that its reflects the population with Diabetic Foot Syndrome. In the literature, we can observe that the percentage of male with Diabetic Foot Syndrome is higher than the percentage of female with Diabetic Foot Syndrome (Navarro-Flores E, Cauli O. Quality of Life in Individuals with Diabetic Foot Syndrome. Endocr Metab Immune Disord Drug Targets. 2020;20(9):1365-72).
We agree with the reviewer's comment about the surgical management of Diabetic Foot Osteomyelitis and the authors have added the following limitation in the discussion section, lines 221-224:
"In addition, another limitation of the study was the difference between the number of patients who received surgical versus medical treatment, with the number of patients managed surgically being higher."
Yes, the patient data included in this study belong to a sample of patients with Diabetic Foot Osteomyelitis, from whom data on diagnosis, treatment and monitoring were collected.
Tardaguila-Garcia A, Garcia-Alvarez Y, Garcia-Morales E, Lopez-Moral M, Sanz-Corbalan I, Lazaro-Martinez JL. Long-Term Complications after Surgical or Medical Treatment of Predominantly Forefoot Diabetic Foot Osteomyelitis: 1 Year Follow Up. J Clin Med. 2021;10(9).
Tardaguila-Garcia A, Garcia Alvarez Y, Garcia-Morales E, Alvaro-Afonso FJ, Sanz-Corbalan I, Lazaro-Martinez JL. Utility of Blood Parameters to Detect Complications during Long-Term Follow-Up in Patients with Diabetic Foot Osteomyelitis. J Clin Med. 2020;9(11).
Tardaguila-Garcia A, Garcia-Alvarez Y, Sanz-Corbalan I, Lopez-Moral M, Molines-Barroso RJ, Lazaro-Martinez JL. Could X-ray Predict Long-term Complications in Patients with Diabetic Foot Osteo-myelitis? Adv Skin Wound Care. 2022;35(8):1-5.

Reviewer 2 Report
The authors present the results of a prospective observational study included 116 patients with DFO. They found that high levels of eosinophils are associated with a longer healing time. No relationship has been found between healing time and different radiological signs of osteomyelitis, clinical signs of infection, other blood parameters, surgical or medical treatment.
The study is interesting, but some revisions are needed.
1) Why the authors define peripheral arterial disease in case of ABI >1.4? Is not sure that all patients with ABI>1.4 are affected by PAD. In some cases this value is related to peripheral autonomic neuropathy and medial arterial calcification.
2) In Table 1 there are some points that are not clear in relation to the diabetic foot epidemiology. May you confirm that the majority of patients have type 1 diabetes? May you check if SD for A1c and duration of ulcer are correct.
3) May you explain how do you manage PAD? PAD patients have been revascularized?
4) Why the authors did not consider other factors such as ulcer size, presence of PAD o ischemia in their analysis? A multivariate analysis, including all these factors, may be useful to know if eosinophils are indipendent factors of healing time.
5) Concerning the current data, Authors should discuss how eosinophils could be related to longer healing time.
Author Response
Thank you very much for your review. Following we are answering concerns that you have detailed in your review. Yellow highlight text indicates the modified or added text in the new version of the manuscript.
Reviewer: 2
Comments and Suggestions for Authors
The authors present the results of a prospective observational study included 116 patients with DFO. They found that high levels of eosinophils are associated with a longer healing time. No relationship has been found between healing time and different radiological signs of osteomyelitis, clinical signs of infection, other blood parameters, surgical or medical treatment.
The study is interesting, but some revisions are needed.
1) Why the authors define peripheral arterial disease in case of ABI >1.4? Is not sure that all patients with ABI>1.4 are affected by PAD. In some cases this value is related to peripheral autonomic neuropathy and medial arterial calcification.
Thank you for your comment. In order to clarify the diagnosis of PAD, the authors have modified the text in the methodology section (lines 82-89):
The vascular examination consisted of palpation of distal pulses (dorsal pedis and posterior tibial); completion of the ankle-brachial index (ABI); completion of the toe-brachial index (TBI); and assessment of transcutaneous oxygen pressure (TcPO2) (TCM4 transcutaneous monitor; Radiometer Medical, Brønshøj, Denmark) (11). A peripheral arterial disease was considered when the patient presented: absence of palpation of distal pulses and ABI <0.9. Also, PAD was considered when the patient presented: absence of palpation of distal pulses and ABI >1.4 (medial arterial calcification), with the TBI <0.7 and TcPO2 <30mmHg (12, 13).
2) In Table 1 there are some points that are not clear in relation to the diabetic foot epidemiology. May you confirm that the majority of patients have type 1 diabetes? May you check if SD for A1c and duration of ulcer are correct.
Thank you very much for your comment. The authors have reviewed table 1 and indeed the frequencies of diabetes type were wrong, with 12 (10.3%) cases with type 1 diabetes and 104 (89.7%) cases with type 2 diabetes. Regarding the data on standard deviation of HbA1c and ulcer duration the authors have reviewed it and the data are correct as shown.
3) May you explain how do you manage PAD? PAD patients have been revascularized?
Thank you for your enquiry. Patients who required revascularisation or had critical limb ischaemia were excluded from the study. To clarify this point, the authors have added the following information in the methodology section (lines 60-63):
Exclusion criteria were patients who required major amputation to treat DFO; patients who required revascularisation and patients with critical limb ischemia according to the IWGDF 2019 guidelines (8); pregnant or lactating women.
- Hinchliffe RJ, Forsythe RO, Apelqvist J, Boyko EJ, Fitridge R, Hong JP, et al. Guidelines on diagnosis, prognosis, and management of peripheral artery disease in patients with foot ulcers and diabetes (IWGDF 2019 update). Diabetes Metab Res Rev. 2020;36 Suppl 1:e3276.
4) Why the authors did not consider other factors such as ulcer size, presence of PAD o ischemia in their analysis? A multivariate analysis, including all these factors, may be useful to know if eosinophils are independent factors of healing time.
Thank you very much for your suggestion. We have not considered the reviewer's proposed analysis in this study, but we will take his interesting recommendation into account for future studies.
5) Concerning the current data, Authors should discuss how eosinophils could be related to longer healing time.
Thank you very much for your recommendation. The authors consider the finding that the increase in eosinophils is related to the increase in healing times to be a casual and isolated finding, finding no justification in the previously published scientific literature. To clarify this point, the authors have included the following information in the discussion section (lines 193-197):
“The authors consider the finding of a relationship between the increase in eosinophils and the increase in time to healing to be a casual and isolated finding, as the other blood inflammatory parameters have not been shown to be related to an increase in healing times, and furthermore there is no data in the scientific literature to justify this finding.”

Reviewer 3 Report
This paper is aimed at evaluating time to healing in patients with DFO and normalization of altered basal biochemical parameters (or at least some of them) .
The paper is interesting although presenting some important points of concern:
1. As suggested by Authors what is lacking is a comparison group, since all patients healed from DFO and the only considered variable was time to healing: a solution could be to categorize patients for instance by healing within or beyond the term of 12 weeks or including a time to event (healing) analysis (K-M or logistic modeling analysis). Moreover I suggest including in the analysis all the patients who presented DFO other than the 116 used in the present analysis using a sort of flow chart, to evaluate the global fate of them: how many were amputated (major and minor) and how many did survive. Finally including and considering comorbidities is essential also in evaluating the eventual normalization of biochemical paraments.
2. It is unclear whether all patients normalized all considered parameters at healing time. In other words did some patents with DFO heal without normalizing all parameters?
3. Why normalization of Hb (correction of eventual anemia) has not been considered?
4. All considered parameters should be included in Table 1, including also parameters such as TcPO2, TBI, or assessment of deep sensitivity, and evaluating all parameters at the last visit. The final prognosis and the change of biochemical parameters must globally consider all of them.
5. Percentage variation is important: for instance what the value in normalization of creatinine, without considering the percent variation from the basal value? And in this case how many patients were affected by more advanced stage kidney failure?
6. How many were treated by revascularization since about half of patients were affected with PAD? And how many patients were treated with minor amputations of bones affected with osteomyelitis?
7. Finally what the hypothesis to corroborate the finding of normalization of eosinophils? In other words could this data only represent a chance finding, without any causal relation. When considering statistically many differences at the same time this is a possible expected explanation.
8. In table 1 the number % of patients with type 1 diabetes should be reversedThis paper is aimed at evaluating time to healing in patients with DFO and normalization of altered basal biochemical parameters (or at least some of them) .
The paper is interesting although presenting some important points of concern:
1. As suggested by Authors what is lacking is a comparison group, since all patients healed from DFO and the only considered variable was time to healing: a solution could be to categorize patients for instance by healing within or beyond the term of 12 weeks or including a time to event (healing) analysis (K-M or logistic modeling analysis). Moreover I suggest including in the analysis all the patients who presented DFO other than the 116 used in the present analysis using a sort of flow chart, to evaluate the global fate of them: how many were amputated (major and minor) and how many did survive. Finally including and considering comorbidities is essential also in evaluating the eventual normalization of biochemical paraments.
2. It is unclear whether all patients normalized all considered parameters at healing time. In other words did some patents with DFO heal without normalizing all parameters?
3. Why normalization of Hb (correction of eventual anemia) has not been considered?
4. All considered parameters should be included in Table 1, including also parameters such as TcPO2, TBI, or assessment of deep sensitivity, and evaluating all parameters at the last visit. The final prognosis and the change of biochemical parameters must globally consider all of them.
5. Percentage variation is important: for instance what the value in normalization of creatinine, without considering the percent variation from the basal value? And in this case how many patients were affected by more advanced stage kidney failure?
6. How many were treated by revascularization since about half of patients were affected with PAD? And how many patients were treated with minor amputations of bones affected with osteomyelitis?
7. Finally what the hypothesis to corroborate the finding of normalization of eosinophils? In other words could this data only represent a chance finding, without any causal relation. When considering statistically many differences at the same time this is a possible expected explanation.
8. In table 1 the number % of patients with type 1 diabetes should be reversed
9. Title could be better introduced without a question format using variables instead of factors
Author Response
Thank you very much for your review. Following we are answering concerns that you have detailed in your review. Yellow highlight text indicates the modified or added text in the new version of the manuscript.
Reviewer: 3
Comments and Suggestions for Authors
This paper is aimed at evaluating time to healing in patients with DFO and normalization of altered basal biochemical parameters (or at least some of them) .
The paper is interesting although presenting some important points of concern:
- As suggested by Authors what is lacking is a comparison group, since all patients healed from DFO and the only considered variable was time to healing: a solution could be to categorize patients for instance by healing within or beyond the term of 12 weeks or including a time to event (healing) analysis (K-M or logistic modeling analysis). Moreover I suggest including in the analysis all the patients who presented DFO other than the 116 used in the present analysis using a sort of flow chart, to evaluate the global fate of them: how many were amputated (major and minor) and how many did survive. Finally including and considering comorbidities is essential also in evaluating the eventual normalization of biochemical paraments.
Thank you very much for your suggestions. As we have reflected in the discussion section, the main limitation of our study is that we did not compare our results with a control group (patients without DFO), so the results are only extrapolated to patients who have had DFO. According to the reviewer's recommendation, the authors have added a survival curve with the probability of healing in patients with elevated eosinophil values. Nevertheless, in future studies, the authors will take into account the analyses proposed by the reviewer, also taking into account the analysis of comorbidities, as well as including patients who have not healed for different reasons, such as patients with major or minor amputation, patients who have died or patients who have not completed treatment for different reasons.
The following text and figure have been added in the methodology section (lines 138-139) and in the results section (lines 159-160):
“The kaplan-Meier test was used to describe the survival of healing time in patients with elevated eosinophil values.”
“Figure 1 shows the Kaplan-Meier survival curve for healing time in patients with elevated eosinophils.”
Figure 1. Kaplan-Meier survival curve for healing time in patients with elevated eosinophils.
- It is unclear whether all patients normalized all considered parameters at healing time. In other words did some patents with DFO heal without normalizing all parameters?
Thank you very much for your comment. Monitoring of analytical parameters has not been considered in this study, as the aim of the study was: To compare the time until healing in patients who suffer from Diabetic Foot Osteomyelitis according to the presence of clinical signs of infection, the increased blood parameters, the presence of different radiological signs and according to the treatment received for the management of Diabetic Foot Osteomyelitis.
In a previous publication of our Research Group, the evolution of blood parameters in the follow-up of patients undergoing Diabetic Foot Osteomyelitis has already been analysed. In this analysis, the patients healed independently of the normalisation or not of the blood parameters: Tardaguila-Garcia A, Garcia Alvarez Y, Garcia-Morales E, Alvaro-Afonso FJ, Sanz-Corbalan I, Lazaro-Martinez JL. Utility of Blood Parameters to Detect Complications during Long-Term Follow-Up in Patients with Diabetic Foot Osteomyelitis. J Clin Med. 2020;9(11).
- Why normalization of Hb (correction of eventual anemia) has not been considered?
Thank you very much for your contribution. The analytical parameters considered in this study have not included the Hb value, as the authors have considered analytical parameters that have been previously described in the literature, such as inflammatory blood parameters. However, we take into account their recommendation for future studies.
- All considered parameters should be included in Table 1, including also parameters such as TcPO2, TBI, or assessment of deep sensitivity, and evaluating all parameters at the last visit. The final prognosis and the change of biochemical parameters must globally consider all of them.
Thank you for the recommendation. According to the suggestion, the authors have added the following data in Table 1: the presence of clinical signs of infection, blood parameter data, signs compatible with Diabetic Foot Osteomyelitis on X-ray, and vascular and neurological examination data.
- Percentage variation is important: for instance what the value in normalization of creatinine, without considering the percent variation from the basal value? And in this case how many patients were affected by more advanced stage kidney failure?
Thank you very much for your comment. In the case of creatinine the authors have considered elevated values in cases where creatinine >1.1 mg/dl, and normal values, according to our laboratory, were set at 0.6-1.1 mg/dl.
The stages of renal failure have not been stratified in this study, but thank you very much for the recommendation, as the authors intend to use this information for the development of future studies, recording creatinine data together with glomerular filtration rate to stratify renal failure.
- How many were treated by revascularization since about half of patients were affected with PAD? And how many patients were treated with minor amputations of bones affected with osteomyelitis?
Thank you for your questions.
Patients who required revascularisation or had critical limb ischaemia were excluded from the study. To clarify this point, the authors have added the following information in the methodology section (lines 60-63):
Exclusion criteria were patients who required major amputation to treat DFO; patients who required revascularisation and patients with critical limb ischemia according to the IWGDF 2019 guidelines (8); pregnant or lactating women.
- Hinchliffe RJ, Forsythe RO, Apelqvist J, Boyko EJ, Fitridge R, Hong JP, et al. Guidelines on diagnosis, prognosis, and management of peripheral artery disease in patients with foot ulcers and diabetes (IWGDF 2019 update). Diabetes Metab Res Rev. 2020;36 Suppl 1:e3276.
There were no patients treated with minor amputations, as surgical patients were managed by conservative surgery as indicated in the methodology section: "There were no patients treated with minor amputations, as surgical patients were managed by conservative surgery as indicated in the methodology section: "All the surgeries were performed by the same surgeon (J.L.L.M.), a specialist in conservative foot surgery, defined as procedures in which only infected bone and non-viable soft tissue are removed, but no amputation of any part of the foot is undertaken (17)".
- Aragon-Sanchez J. Treatment of diabetic foot osteomyelitis: A surgical critique. Int J Low Extrem Wounds. 2010;9(1):37-59.
- Finally what the hypothesis to corroborate the finding of normalization of eosinophils? In other words could this data only represent a chance finding, without any causal relation. When considering statistically many differences at the same time this is a possible expected explanation.
Thank you very much for your recommendation. The authors consider the finding that the increase in eosinophils is related to the increase in healing times to be a casual and isolated finding, finding no justification in the previously published scientific literature. To clarify this point, the authors have included the following information in the discussion section (lines 193-197):
“The authors consider the finding of a relationship between the increase in eosinophils and the increase in time to healing to be a casual and isolated finding, as the other blood inflammatory parameters have not been shown to be related to an increase in healing times, and furthermore there is no data in the scientific literature to justify this finding.”
- In table 1 the number % of patients with type 1 diabetes should be reversed
Thank you very much for your comment. The authors have reviewed table 1 and indeed the frequencies of diabetes type were wrong, with 12 (10.3%) cases with type 1 diabetes and 104 (89.7%) cases with type 2 diabetes.
- Title could be better introduced without a question format using variables instead of factors
Thank you very much for your suggestion, we have modified the title (lines 2-3) accordingly: “Variables that could influence healing time in patients with diabetic foot osteomyelitis”.

Reviewer 4 Report
Methods - how was sample size determined?
line 61 - why only increased blood parameters? why not stated as outside of reference range in case of lower values e.g low albumin. Even if WBC count typically elevated, this cannot be assumed and if low, could be relevant too
line 95-6 - extensive loss of what?
line 98 - at what stage and how many times were blood samples taken for analysis?
line 129-30 - states 'the U- Mann Whitney t-test was used' but a t-test is not the same as the Mann-Whitney test
Results:line 134 - the +/- is missing after the ulcer healing time
Discussion
line 185 should read ' there is NOT enough evidence'
states that blood samples were taken after antibiotics started, but not at what stage and how many times?
a paragraph signposting the potential relevance of elevated eosinophils, both within the context of metabolic disease and for the inflammatory status of the population concerned should be included - does this reflect the worse diabetic disease for pts with poor healing (was there correlation between glycemic control and eosinophil levels), or a sub-group of DFO patients who have a separate underlying factor related to prolonged healing and eosinophils levels
Author Response
Thank you very much for your review. Following we are answering concerns that you have detailed in your review. Yellow highlight text indicates the modified or added text in the new version of the manuscript.
Reviewer: 4
Comments and Suggestions for Authors
Methods - how was sample size determined?
Thank you very much for your comment. Yes, the authors performed a sample size calculation using the GRANMO® programme. Authors have added the following information in the methodology section (lines 140-142): “The sample size was calculated using GRANMO®, estimating a required sample of 80 subjects (study power-80.0%, type 1 error-5.0% and predictable loss to follow-up-15.0%).”
line 61 - why only increased blood parameters? why not stated as outside of reference range in case of lower values e.g low albumin. Even if WBC count typically elevated, this cannot be assumed and if low, could be relevant too
Thank you for your comment. In the sample recruited in this study, none of the patients showed decreased values of blood parameters, therefore no such comparisons could be made. For this reason, the authors considered analysing the elevated blood parameters and their possible influence on healing times, expecting that the higher the elevation, the worse the infection control and therefore the longer the time to healing, although the results have shown that this is not the case.
line 95-6 - extensive loss of what?
Thank you for the question. The authors have checked the text and there was an error, the paragraph has been modified (lines 96-103) “On the other hand, the signs of infection proposed by the Infectious Diseases Society of America (IDSA) guidelines (15) were also assessed: evidence of systemic inflammatory response, rapid progression of infection, extensive necrosis or gangrene, crepitus on examination or soft tissue gas on imaging, extensive ecchymosis or petechiae, especially haemorrhagic bullae, new onset wound, pain out of proportion to clinical findings, the recent loss of neurological function, critical limb ischaemia, extensive soft tissue loss, extensive bony destruction especially in midfoot and hindfoot, and failure of infection to improve with appropriate therapy.”
- Lipsky BA, Berendt AR, Cornia PB, Pile JC, Peters EJ, Armstrong DG, et al. 2012 Infectious Diseases Society of America clinical practice guideline for the diagnosis and treatment of diabetic foot infections. Clin Infect Dis. 2012;54(12):e132-73.
line 98 - at what stage and how many times were blood samples taken for analysis?
Thank you for your question. In accordance with your suggestion and for clarification, the authors have added the following text in the methodology section (lines 104-105): “Laboratory blood tests were performed in all patients, within 24 hours or up to a maximum of 48 hours after diagnosis of DFO”.
line 129-30 - states 'the U- Mann Whitney t-test was used' but a t-test is not the same as the Mann-Whitney test
Thank you very much, it was a typo that has already been corrected (lines 137-138): “For the comparison of quantitative variables between the two groups, the U-Mann Whitney test was used”
Results:line 134 - the +/- is missing after the ulcer healing time
Thank you very much, the typo has been corrected.
Discussion
line 185 should read ' there is NOT enough evidence'
Thank you very much, the typo has been corrected.
states that blood samples were taken after antibiotics started, but not at what stage and how many times?
Thank you for your question. In accordance with your suggestion and for clarification, the authors have added the following text in the methodology section (lines 104-105): “Laboratory blood tests were performed in all patients, within 24 hours or up to a maximum of 48 hours after diagnosis of DFO”.
a paragraph signposting the potential relevance of elevated eosinophils, both within the context of metabolic disease and for the inflammatory status of the population concerned should be included - does this reflect the worse diabetic disease for pts with poor healing (was there correlation between glycemic control and eosinophil levels), or a sub-group of DFO patients who have a separate underlying factor related to prolonged healing and eosinophils levels
Thank you very much for your recommendation. The authors consider the finding that the increase in eosinophils is related to the increase in healing times to be a casual and isolated finding, finding no justification in the previously published scientific literature. To clarify this point, the authors have included the following information in the discussion section (lines 193-197):
“The authors consider the finding of a relationship between the increase in eosinophils and the increase in time to healing to be a casual and isolated finding, as the other blood inflammatory parameters have not been shown to be related to an increase in healing times, and furthermore there is no data in the scientific literature to justify this finding.”
Regarding poor metabolic control, the authors performed a Person correlation analysis between glycaemia and eosinophils (r= -0.154; p= 0.103) and HbA1c and eosinophils (r= 0.019; p= 0.838). The correlation is low and not statistically significant; therefore, it cannot be associated that poorer metabolic control is reflected in increased eosinophils.

Round 2
Reviewer 2 Report
Thank you for the corrections.
You should include in the study limitations that you did not perform a multivariate analysis.
Author Response
Thank you for your review. Green highlight text indicates the modified or added text in the new version of the manuscript.
Reviewer: 2
Comments and Suggestions for Authors
Thank you for the corrections.
You should include in the study limitations that you did not perform a multivariate analysis.
Thank you for your review. In accordance with your comment, authors have added the following information in the discussion section (lines 219-220):
“In the study, multivariate analysis, including other risk factors such as ulcer size or the presence of ischaemia, has not been performed.”
Reviewer 3 Report
The paper has now greatly improved: only a minor remark: KM is aimed time to healing of osteomyelitis. The analysis is uuslly used to compare time to events of two or more cohorts. In this case for instance patients with lower to those with higher quartile of eosinophils at baseline
Author Response
Thank you for your review. Green highlight text indicates the modified or added text in the new version of the manuscript.
Reviewer: 3
Comments and Suggestions for Authors
The paper has now greatly improved: only a minor remark: KM is aimed time to healing of osteomyelitis. The analysis is uuslly used to compare time to events of two or more cohorts. In this case for instance patients with lower to those with higher quartile of eosinophils at baseline
Thank you very much for your comment. According to your suggestion the authors have modified the survival graphic in the results section (lines 153-163).
“Figure 1 shows the Kaplan-Meier survival curve for healing time in patients with elevated eosinophils p=0.005.
Figure 1. Kaplan-Meier survival curve for healing time in patients with elevated eosinophils.”
